# Service Process Problem-Solving Based on Flow Trimming

**Bai Zhonghang [1,2], Lin Siyue [1,2] and Zhang Xu [2,\***

1   College of Architecture and Art Design, Hebei University of Technology, Tianjin 300130, China
2   National Technological Innovation Method and Tool Engineering Research Center,
    Hebei University of Technology, Tianjin 300401, China
\*   Correspondence: zhangxu@hebut.edu.cn

**Abstract:** Since entering the era of the experience economy, consumers' attention gradually turns toward gaining a pleasant service process experience. This study addresses the service process problem, aiming to discover the root cause of the service process problem and solve it by analyzing the service touchpoints flow delivery process. A method for solving problems in the service process based on flow trimming is proposed. The trimming method and the stochastic dominance rule are applied to the field of service design, which provides new concepts for service process problem solving. The flow is taken as the entry point of the proposed method. First, a flow model of problematic service touchpoints is constructed based on the service blueprint method to visualize the flow delivery process. Then, service process trimming rules are proposed and used as guidance to trim flow disadvantages, and resource analysis is employed to obtain specific programs. Finally, the stochastic dominance rule is used to rank the programs and select the optimal program. Problem solving in the medical treatment service process was taken as an example to trim the fundamental flow disadvantages of problematic service touchpoints. A series of programs were obtained and the optimal program was selected for presentation based on the stochastic dominance rule, which verified the feasibility of the proposed method.

**Keywords:** TRIZ; trimming; flow analysis; service process; stochastic dominance rule

## 1. Introduction

With the continuous growth of services in today's organizations and economies, the importance of understanding the concepts and practices of service innovation is increasing [1]. Service design aims to innovate services, create high-quality services, and provide users with a good experience [2]. Service innovation has many modes, such as service products, service processes, and business model innovation [3] among which process innovation plays a key role in service innovation [4]. Although many studies on service innovation have focused on the innovation of service products, it has been long recognized that the simultaneity of service production and delivery means that service innovation may be the result of service process innovation [5]. Solving the problems of the innovation process is the key to innovative design [6]. However, determining how to use domain logic to analyze the problem, understand the nature of the problem, and make breakthroughs in quality improvement and measurement remain the key challenges of service design [7].

Service touchpoints are the main key points in the service process [8]. The interaction between users and service providers is called service touch, the nodes of which are service touchpoints. User experience mainly depends on these touchpoints throughout the service interaction process [9]. A service is a system that integrates all online and offline resources, as well as front-end and back-end resources, to serve users. Among them, materials, information, and funds reflect the dynamic delivery of services via flow and change between stakeholders [10]. It can be seen from the above that service touchpoints ultimately appear via the interaction between users and service providers, and the formation process can be regarded as a flow transfer process that occurs between stakeholders. For many

process problems, especially regarding technical processes involving material attributes and information changes, it is easier to find the key points of the problem from the flow [11]. At present, the analysis and resolution of problems in the service process mainly revolve around the interaction between users and service providers when they receive services, resulting in a situation in which only the surface problems are solved, and the deeper reasons for the problems are not explored. Therefore, the solution to problems in the service process can begin from the perspective of service touchpoint flow. Traditional flow analysis lacks practical methods by which to model the elements existing in service touchpoints and systematic methods by which to generate different types of innovative programs.

TRIZ is a complete innovation theory proposed by Altshuller et al. after analyzing a large number of high-level patent studies around the world. Li S.P. [12] used TRIZ for patent review and the novel design of vehicle classification systems. Ng Poh Kiat [13] and others conducted a conceptual development of an automatic roof tile transportation system based on TRIZ inspiration, which reduced manual and material handling, reducing the risk of occupational musculoskeletal disorders. Tien-Lun Liu [14] applied natural language processing and TRIZ evolutionary trends to provide applicable patent advice regarding product design for design assistance purposes so as to improve the product development process and problem solving. Spreafico, C. [15] quantified the benefits of TRIZ in sustainability through life cycle assessment. These results, and the subsequent discussion, provide a data history for designers wishing to use TRIZ in ecological design. The main components of the TRIZ theory system include conflict resolution theory and invention principles. Matter-field analysis TRIZ theory includes conflict resolution theory, invention principles, substance-field analysis, standard solution, trimming, etc. Trimming is an important problem-solving tool in TRIZ theory. Trimming is the process of eliminating excessive, harmful, and insufficient functions, while at the same time finding available resources inside and outside the system to perform useful functions, in addition to finally achieving a higher degree of idealization of the system. It also simplifies structures, reduces costs, and enables innovation [16]. The trimming method was initially applied in the product innovation phase and as the product became more mature, it also gradually moved to the process innovation domain.

Darrel [17] combined functional trimming with DFMA and analyzed the optimal entry period of trimming in the life cycle of the technical system. Li [18] proposed an integrated process focusing on process flow and product innovation, and the process trimming method was used to solve key problems, which significantly reduced manufacturing defects and service costs. Mitchell [19] used cutting methods to solve the problem of emulsion damage to products and eliminate the product maintenance process. Bai Zhonghang [20] proposed the trimming process model of multilevel system resource derivation assisted by a standard solution, which improved the operability of useful function reallocation. Yu Fei [21] proposed a process model of trimming innovation design based on the trimming method set, which eliminated system conflicts in the application process and met the design objectives. Tan Zhen [22] applied process trimming to process improvement and proposed a set of perfect process trimming rules, which effectively assisted designers in process improvement and process innovation. Chen Xingzhi [23] established a graphical process trimming function model, trimming rules, and trimming plans, and established an efficient and systematic trimming process. Tan Ruoshi [24] proposed a process problem-solving model based on trimming, which is conducive to improving the efficiency of process problem solving in the future. Zhang Huangao [25] established a product platform evolution process model based on functional trimming and realized the evolution of the ear seedling sort platform. Zhang Wei [26] proposed a process model of trimming innovation design based on the trimming principle set, which effectively made up the limitations of the subjective analysis process when allocating useful functions. Shao Jingfeng [27] improved the efficiency of the process operation of brand logo redesign as a service through the theory and method of service design. Tan Wei [28] proposed a time-improvement manufacturing service process optimization method with a QoS guarantee in order to improve the overall execution efficiency

of the service process and the overall execution efficiency to improve. Shen Weiqun [29] used Lean Six Sigma tools to optimize the library service process and model from both macro- and microperspectives to optimize the library service process and model, aiming to eliminate the defects in the process and implement scientific management of the library. Zhang Qing [30] posited the idea and method of service design in rural tourism service and optimized the rural tourism service process by analyzing the pain and opportunity points, including improving the service touchpoints. Few current studies have combined trimming with service process optimization, and the current service process research lacks universal and systematic solutions. Most studies only study service touchpoints from the single perspective of service consumption without considering the problem from a global perspective, which may lead to ignoring the essence of the problem.

Therefore, this paper proposes to study the service process problem from the perspective of flow analysis, and the specific questions to be addressed include: (1) How to understand and describe the flow analysis in the service process? (2) How to identify and define the flow defects in the service process? (3) How to solve the defects in the service process through TRIZ tools? To address these questions, this paper proposes a TRIZ trimming-based service process problem-solving method. The method starts with flow analysis and modeling based on the service blueprint to visualize the elements of the service contact formation process, with the aim of discovering the fundamental flow defects in the service process. Then, the fundamental flow defects are trimmed by applying the trimming rules to form multiple innovative solutions. Finally, the optimal solution is extracted by using the stochastic dominance criterion. This paper expands the scope of TRIZ theory and combines flow analysis and the trimming method in the service design field, which makes up for the shortcomings of flow analysis and forms a trimming method for the service process, providing a new idea for service design and service innovation. It also has a certain reference value for the subsequent related research. In practice, the flow trimming method for service innovation is designed for the current situation of medical consultation flow problems, which provides a practical reference for solving consultation flow problems in related fields.

## 2. Related Works

### 2.1. Service Touchpoint Flow Delivery Process

The service process is the process in which the service provider delivers the service content to the service receiver (user) in response to the needs of the service receiver [9]. It consists of service touchpoints that interact before, during, and after the service [8]. By experiencing different touchpoints, users can feel the value of the whole service process [31]. Therefore, scholars have mostly solved problems in the service process at the present stage from the perspective of service touchpoints [32,33].

With the increasing intensification of market competition and the diversification of customer needs, the business model has changed from the traditional single operation to a multifaceted collaborative business model with suppliers, customers, and partners [34], People are beginning to realize that service touchpoints are formed by the participation of multirole stakeholders. To solve related problems, it is necessary to determine the relationships between these stakeholders and the flow of resources between them. Baltacioglu [35] defined the service supply chain as a network composed of suppliers, service providers, users, and other support units to complete the resource transactions required to produce services. These resources are transformed into a core of supporting services, which are finally passed on to the user. Service design is a design activity cocreated by multiple stakeholders with the user as the main perspective [36]. From a service perspective, co-creation describes the characteristics of the service, i.e., the service provider and the user create value together. Without the resources provided by the user, the service cannot be carried out [37]. Therefore, service activities do not depend only on the service provider; they also require the participation of the users and all stakeholders [38]. The delivery process of the service touchpoint flow is a complete closed loop, i.e., the user puts forward a demand,

and the stakeholder transforms the demand into a service to meet the demand, as shown in Figure 1.

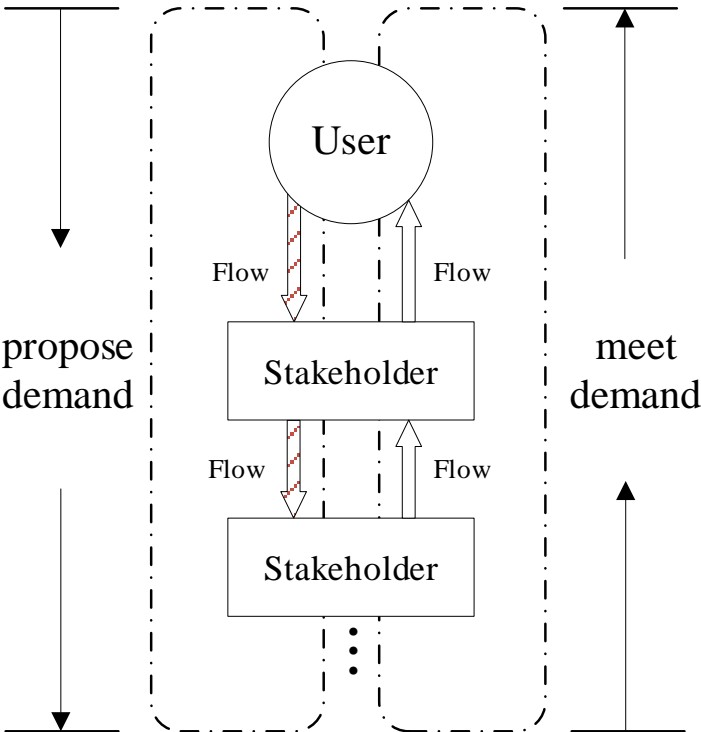

**Figure 1.** The delivery process of the service touchpoint flow.

This paper primarily investigates the delivery process of the service touchpoint flow, employs flow analysis to determine the delivery process, and identifies the root causes of problems in the service process.

### 2.2. Flow Analysis

Flow analysis is a dynamic analysis of the system from a new perspective. It is a development and supplement to the physical field analysis and functional analysis and has a similar process and approach to the functional analysis. It is to find the defects of the flow in the technical system and make up for the deficiencies of the functional analysis [39].

The flow is divided into two categories, namely flow partition disadvantages and operational flow disadvantages, as shown in Figure 2. The former focuses on the effects caused by the flow and can be further subdivided into harmful flow, wasted flow, and inefficient flow. The latter focuses on the problems of the flow itself and can be further subdivided into utilization disadvantage flow and conductivity disadvantage flow. Liu Zhenhui analyzed the flow-based harmful function generation process, established a function-effect-state model and its symbolic system suitable for the study of harmful functions, and constructed a harmful function derivation process model on this basis.

Chen [23] employed flow analysis to identify the flow disadvantages between the job provider and the target object in the system to ultimately locate problematic job activities. Xu [40] studied the product innovation process from the perspective of an expert system and proposed an auxiliary innovation design model based on flow analysis. At present, flow analysis is primarily used in the problem analysis stage, and it is lacking in the promotion of problem solving and the generation of creative programs. Flow analysis is used to sort out the elements in the service touchpoint, refine the problems in the service process, build the entire flow delivery process framework, and complement the use of trimming methods to solve the problems.

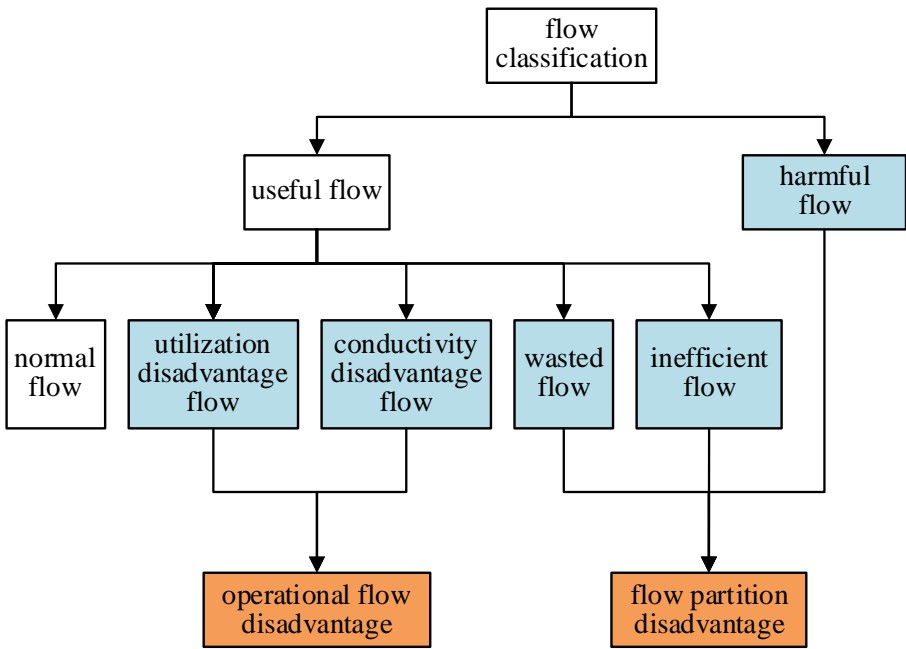

**Figure 2.** Flow disadvantage classification.

*2.3. Process Trimming*

Process trimming (Trimming for the process) is a problem-solving tool extended from the trimming tool in TRIZ theory [23,40]. Process trimming is the process of discovering problematic operations via functional analysis, trimming these problematic operations, and redistributing their useful functions. It is generally divided into four stages, namely the process identification stage, process analysis stage, process trimming stage, and concept program phase.

The process identification stage: The process system that requires improvement is determined, the purpose of which is to solve the technical problems in the process system.

Process analysis stage: The functional analysis of the process that requires improvement is performed to build a functional model, after which the operations to be trimmed are determined. Functional analysis is the basis of the application of the trimming method. In the design process, the reduction of cost or complexity is often the result of both functional analysis and trimming. In process functional analysis, each operation in the process, the contribution of each operation function, the function type, and the performance level are described to clearly reveal the running status of the system and the function performed [41].

Process trimming stage: The useful function types of the operations to be trimmed are judged, and the corresponding process trimming rules are selected to implement trimming.

Concept program stage: Other tools in TRIZ theory are used to complete useful function configuration and present the final plan.

Li [18] proposed an integrated approach for the innovation of technological processes and products based on trimming. Mitchell [19] used the trimming tool to trim the latex coating process, and finally used a simply styled suction nozzle and transfer tube to replace the original complex process. Chen and Xu [23] established a graphical process trimming functional model, trimming rules, and trimming plan with the goal of creating an efficient and systematic trimming process. Tan [24] researched and analyzed a process in the process industry, proposed a process problem-solving method based on trimming, and established a process problem-solving model. According to these previous studies, it is evident that the TRIZ trimming method is of great significance in the research field of process problems. However, the current research focuses on the trimming of the product production process, which is different from the service process. Therefore, the existing process trimming methods are not fully applicable to the service process. Thus, in this

paper, the trimming method is used to trim the flow disadvantages of the service process, and a trimming method suitable for the service process is formed. In the process analysis stage, flow analysis is used instead of functional analysis, and trimming rules suitable for the service process are proposed for the process trimming stage.

## 3. Service Process Trimming Method

In service systems, the relational organization of different stakeholders will constitute an impact on the design of service touchpoints as well as on the overall service system, making it possible to form different service strategies [10]. Flow analysis allows visualizing the interactions between stakeholders who undertake the production, delivery, and use, and represents the flow of material, information, and money between them. Therefore, service process trimming uses flow as an entry point, first identifying problematic service touchpoints, then performing flow analysis on the flow delivery process of the problematic service touchpoints, and finally identifying flow defects and tailoring them. Therefore, service process trimming uses flow as the entry point, first identifying problematic service touchpoints, then performing flow analysis on the flow delivery process of problematic service touchpoints, and finally discovering flow defects and performing trimming.

### 3.1. Flow Analysis Based on the Service Blueprint Method

Flow analysis is based on the material, financial, and information flows between stakeholders involved in problematic service touchpoints and aims to discover flow defects in the service process. In this study, flow analysis is applied to the service process trimming method to lay the foundation for implementing trimming instead of functional analysis.

### 3.1.1. Stakeholder Stratification Based on the Service Blueprint

The service process covers a complex and variable set of stakeholders, and the results of stakeholder classification vary according to the basis, dimensions, and application domains of the classification [42]. To sort out the interactions between stakeholders in the service process, it is necessary to classify them via unified stratification.

The service blueprint method is a process-based service design method [43]. There are three special lines in the service blueprint, namely the service interaction line, the visible line, and the internal interaction line [44]. These three lines gradually decompose the work of various functional departments involved in the service provision process, namely customer action, front-end interaction, back-end interaction, and the support process. As a whole, the service blueprint reflects coordinated activities and their mutual relationships with various elements within the same time node [45].

According to the three dividing lines in the service blueprint, the stakeholders in the flow delivery process of service touchpoints can be divided into four levels, namely users, stakeholders in the system who are in direct touch with users, stakeholders in the system who are not in direct touch with users, and support systems/collaborators outside the system. More intuitive diagrammatic language is used to describe these users, as shown in Figure 3.

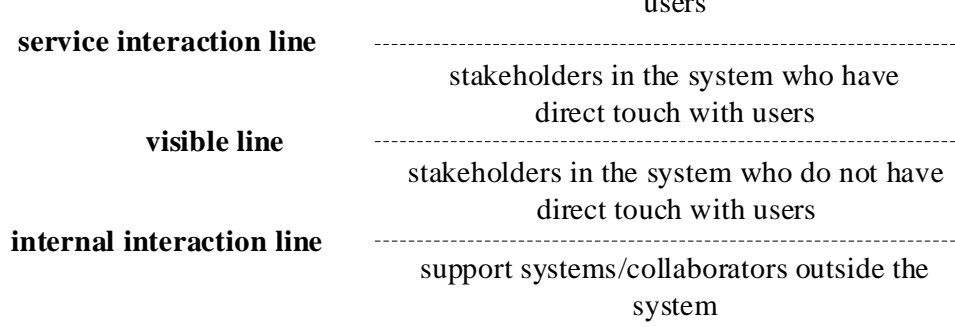

**Figure 3.** The stakeholders in the flow delivery process of service touchpoints.

### 3.1.2. Classification of Flow Disadvantages in the Service Process

Any product innovation problem is closely related to the flow, attributes, and harmful functions generated by the interaction of the technical system of the problem [46]. The same is true for service design. Let the party that provides the flow among the four stakeholders be called the flow provider and let the party that receives the flow be called the flow receiver. The four types of stakeholders can switch between flow providers and flow receivers. The flow disadvantage classification method proposed by Liu [47] is utilized in the present study to provide specific descriptions and examples of various flow defects in combination with the characteristics of the service process, as reported in Table 1.

**Table 1.** Classification of flow disadvantages in the service process.

| Type of Flow Disadvantage | Meaning | Content | Example |
|---|---|---|---|
| Harmful flow | The flow provider provides a useful flow to the flow receiver while providing harmful flow. | Flow self-damage; flow damage channel; flow damage other objects | Some web pages will provide users with useful information flow, but will simultaneously push some vulgar content, i.e., harmful information flow. |
| Inefficient flow | The flow provider provides a useful flow to the flow receiver, but the flow is too small or too little. | Inefficient useful material; inefficient useful information; inefficient useful funds | A hospital provides an inefficient number of beds, and patients need to wait for admission. |
| Wasted flow | The flow provider provides a useful flow to the flow receiver, but the flow is too large or too much. | Wasted useful material; wasted useful information; wasted useful funds | Faced with a lot of similar advertising information, readers are confused and face difficulties in choosing information, and there is an overwhelming amount of useful information. |
| Utilization disadvantage flow | The flow provider provides a useful flow to the flow receiver, but the flow is not properly utilized. There are two situations: over-utilization and under-utilization. | Grey zones; channel damages flow; other objects damage flow; flow damages channel; flow damages itself; flow damages other objects | During the Spring Festival travel season, there are a lot of travelers, and the trains are over-utilized, causing congestion in the carriages. |
| Conductivity disadvantage flow | The flow provider provides a useful flow to the flow receiver, but the flow itself has problems that affect the efficiency of the circulation. | Bottlenecks; stagnant zones; recirculation zones; poorly transferable flow; long flow channel; high channel resistance; low flow density; a large number of transformations | During periods of heavy traffic, the resistance at the crossroads increases significantly, causing congestion. |

### 3.1.3. Model Construction and Problem Excavation of Service Touchpoint Flow

In this paper, the four types of stakeholders involved in the problematic service touchpoints are graphically illustrated, and their interrelationships are reflected via images, i.e., the delivery of the service touchpoint flow is presented in the form of a flow model. The sequence of building the flow model is as follows:

(1) Stakeholder recognition of problematic service touchpoints based on the service blueprint. The four types of stakeholders involved in the problematic service touchpoints are identified by their level.

(2) Flow recognition and flow model construction. According to the stakeholders identified in step (1), the type of flow between two adjacent types of stakeholders is identified, and the flow model of the problematic service touchpoint is drawn, as shown in Figure 4.

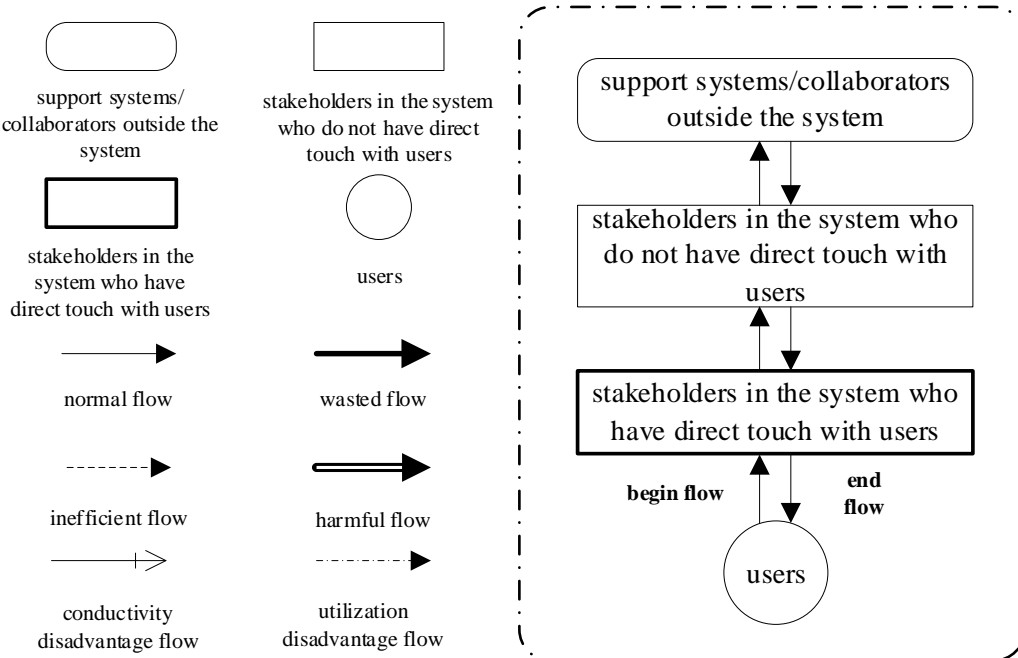

**Figure 4.** The flow model of service touchpoints.

To solve the problem innovatively, it is necessary to first clarify the problem, that is, to clarify the flow of the implementation of the trimming method. The basic principle is that each cause of the underlying flow disadvantage is directly related to the attribute of the flow provider or the flow itself. First, the flow disadvantage that causes user dissatisfaction is identified according to the flow model, after which the uppermost flow disadvantage related to it is identified and defined as the final flow to be trimmed.

### 3.2. Service Process Trimming

Service process trimming solves problems in the process by removing flow defects in the system that ultimately lead to service touchpoint problems in meeting user needs.

#### 3.2.1. Service Process Trimming Rules

Flow refers to the occurrence between the components and connects the components to form an object with certain functions, goals, and structures, as well as flow and transfer characteristics. Service process trimming is essentially the trimming of flow disadvantages. However, after the flow disadvantages are trimmed, the useful functions provided by the useful flow are also trimmed, so the useful functions must be redistributed. Trimming generally redistributes useful functions from the three aspects of deletion, replacement, and addition [48]. Based on this, the following questions are proposed to guide the trimming of the service process.

Rule 1: Can flow disadvantages be deleted directly?
Rule 2: Can flow disadvantage receivers be deleted directly?
Rule 3: Are there other resources within or outside the system that can replace the flow?
Rule 4: Are there other resources within or outside the system that can replace the flow provider to provide useful flows?
Rule 5: Are there other resources within or outside the system that can replace the flow receiver to accept useful flows?
Rule 6: Is it possible to eliminate flow disadvantages by adding other stakeholders?

#### 3.2.2. Service Process Trimming Process Model

The method described in the previous section is used to construct a process model of service process trimming, as shown in Figure 5.



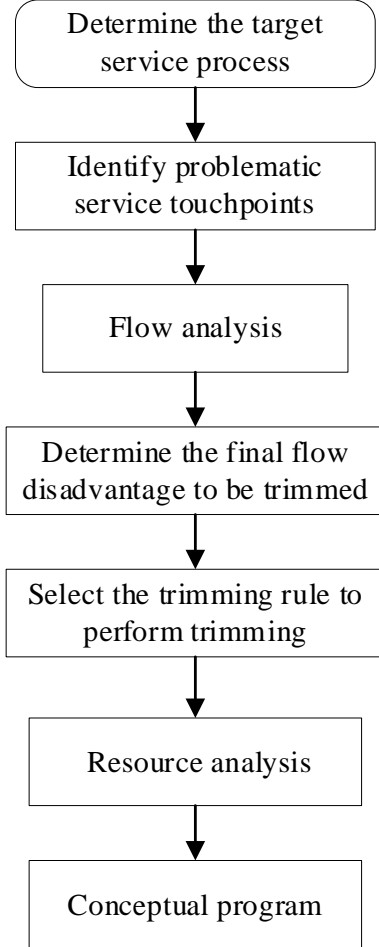

**Figure 5.** The process model of service process trimming.

(1)  Determine the target service process. Based on experience and data, the problematic service process to be analyzed is determined.

(2)  Identify problematic service touchpoints. There are many ways to determine problematic service touchpoints, such as via the use of user journey maps, field surveys, etc.

(3)  Flow analysis. Identify the four types of stakeholders involved in the problem service touchpoints. Then, identify the types of flows provided and build a stakeholder flow model.

(4)  Determine the final flow disadvantage to be trimmed. Each service touchpoint contains multiple flows. The problems generated by the service touchpoints are ultimately presented through flow disadvantages between users and stakeholders in the system who have direct contact with users. Therefore, to find the source of the problem, it is necessary to identify the initial flow disadvantage, i.e., the final flow disadvantage to be trimmed.

(5)  Select the trimming rule to perform trimming. According to the type of flow disadvantage, various trimming rules are attempted for trimming.

(6)  Performing resource analysis to achieve concept solving. In a way, the process of trimming is the process of using and reconfiguring resources, and the method of resource analysis can help find effective resources. In M-TRIZ theory [39], resources are divided into material, social, information, time, and space resources. In this paper, M-TRIZ resource analysis is utilized to find available resources to convert the conceptual program into a concrete program.

## 4. Optimal Program Selection Based on The Stochastic Dominance Rule

Considering the problems in the service touchpoints, the use of different trimming rules for trimming may produce a variety of preliminary programs. However, it is often only necessary to select the program that best meets the user's needs for a detailed presentation.

The stochastic dominance criterion is a decision rule that uses partial information to form a biased order. [49]. Its main feature is that it does not require too many strict assumptions, and it can yield accurate alternative ranking results. This method is widely used in the fields of satisfaction evaluation [50] and program optimization [51]. The application of the stochastic dominance rule to the selection of the optimal program for service design can clarify the comprehensive ranking of programs and yield a relatively optimal program. First, we obtain information on the distribution of the evaluation value of users' satisfaction with each tailoring solution under different evaluation dimensions through a questionnaire; Then, based on the stochastic dominance rule, the stochastic dominance relation between the solutions of the two programs in a certain evaluation dimension is determined. Finally, we apply the PROMETHEE II method to rank the solutions under each evaluation dimension and rank the solutions comprehensively to determine the optimal solution.

### 4.1. The Distribution Function of the Evaluation Values of a Program in Different Evaluation Dimensions

For the convenience of analysis, some sets and quantities involved in this section are subsequently defined.

$C = \{C_1, C_2, \ldots, C_n\}$: The set of evaluation dimensions, where $C_j$ represents the $j$-th evaluation dimension, $j \in n$.

$S = \{S_1, S_2, \ldots, S_n\}$: The weight vector of the evaluation dimension, where $S_j$ represents the weight or importance of the evaluation dimension $C_j$, $\sum_{j=1}^{n} S_j = 1$. The weight vector of the evaluation dimension can be given directly by decision-makers or obtained by a questionnaire survey.

The problematic service touchpoints in the service process are denoted as $a, b, c \ldots$

$a_k = \{a_1, a_2, \ldots, a_m\}$: The set of programs for the problematic service touchpoint $a$, where $a_k$ represents the $k$-th program, $k \in m$.

$W_{jk} = (W_{j1}, W_{j2}, \ldots W_{jm})$: The evaluation value vector of the user's satisfaction with each program corresponding to the evaluation dimension $C_j$, where $W_{jk}$ represents the evaluation value of the user's satisfaction with the program $a_{jk}$, corresponding to the evaluation dimension $C_j$. To obtain the evaluation values of users' satisfaction with different programs in a certain evaluation dimension, a questionnaire survey is required. Via the use of the Likert five-point scale, different scores reflect users' satisfaction with different programs.

It is considered that $W_{jk}$ is a discrete stochastic variable within the interval $[a,b]$. When $W_{jk}$ is a discrete stochastic variable, the distribution law can be expressed as $P_{jk}\left(W_{jk}\right) = \rho_{jk}^s$. The pulse function $\delta_{jk}(W)$ can be used to define its probability density function, namely $f_{jk}(w) = \sum_{o=1}^{o'} \delta_{jk}\left(w - w_{jk}^o\right)$, where $w_{jk}^o$ represents the discrete value, i.e., the user's evaluation value of the program $a_k$, $o'$ represents the number of discrete values, and $\rho_{jk}^o$ represents the probability of discrete values, i.e., the probability that the evaluation value given by the user to the program $a_k$ is $w_{jk'}^o$, which satisfies $\rho_{jk}^o \geq 0$ and $\sum_{o=1}^{o'} \rho_{jk}^o = 1$. The cumulative distribution function can be expressed as $F_{jk}(w) = \sum_{W \geq a} f_{jk}(w)$, and its mathematical expectation can be expressed as $u_{jk} = \sum_{o=1}^{o'} w_{jk}^o \rho_{jk}^o$

*4.2. Determination of the Stochastic Dominance Relation of Pairwise Programs in a Certain Evaluation Dimension*

The stochastic dominance relation of pairwise programs in a certain dimension can be judged by the distribution function relationship of the vector of the evaluation value of the user's satisfaction with the program of the pairwise programs.

According to the stochastic dominance rule [44] and the distribution function of pairwise programs, the stochastic dominance relation matrix is established, and is denoted as $H = \left[h_{kk'}^{j}\right]_{m \times m}$, where $h_{kk'}^{j}$ represents the stochastic dominance relation between the programs $a_k$ and $a_{k'}$ in the evaluation dimension $C_j$.

$$h_{kk'}^{j} = \begin{cases} SD, F_{jk}(W)SDF_{jk'}(W) \\ -, others \end{cases}, \ (k, k' \in m; k \neq k'; j \in n) \tag{1}$$

where $F_{jk}(W)SDF_{jk'}(W)$ means that $F_{jk}(W)$ is stochastically more dominant than $F_{jk}(W)$, i.e., the evaluation value of program $a_k$ is stochastically better than that of program $a_{k'}$ in the evaluation dimension, which is denoted as $h_{kk'}^{j} = SD$. If there is no stochastic dominance relation between the evaluation values of the programs $a_k$ and $a_{k'}$, it is denoted as $h_{kk'}^{j} = -$.

*4.3. Program Ranking*

The PROMETHEE II method [45] is a complete ranking method based on the ranking priority relationship. In this method, the priority degree function is used to describe the degree of dominance between the programs for a certain evaluation dimension, after which the overall priority degree matrix of the comparison of the two programs is established by judging the degrees of superiority and inferiority between them. According to the overall priority degree matrix, the programs can be comprehensively ranked by calculating the "outflow" and "inflow" of each program and the ranking value [46].

4.3.1. Calculation of the Overall Priority Degree Function of Each Program

For the evaluation dimension $C_j$, there are three relationships between any two programs $a_k$ and $a_{k'}$:

(1) The program $a_k$ is strictly dominant over the program $a_{k'}$. At this time,
$$F_{jk}(W)SDF_{jk}(W) \ and \ u_{jk} \geq u_{jk'} + \varepsilon_j$$
(2) The program $a_k$ is weakly dominant over the program $a_{k'}$. At this time,
$$F_{jk}(W)SDF_{jk}(W) \ and \ u_{jk'} < u_{jk} < u_{jk'} + \varepsilon_j \tag{2}$$
(3) The program $a_k$ is not dominant over the program $a_{k'}$. There is no
$$F_{jk}(W)SDF_{jk}(W) \ at \ this \ time.$$

where $\varepsilon_j$ represents the preference threshold value of the evaluation dimension $C_j$ and is related to the expected difference of the evaluation values of the pairwise programs. Here, the average value of the expected difference of the evaluation values of the pairwise programs is taken as the preference threshold [43], the calculation formula for:

$$\varepsilon_j = \frac{2}{m(m-1)} \sum_{k=1}^{} \sum_{k'=1; k \neq k'}^{} d_{kk'}^{j} (j \in n) \tag{3}$$

where $d_{kk'}^{j}$ represents the expected absolute difference between the pairwise programs for the evaluation dimension $C_j$, as follows.

$$d_{kk'}^{j} = \begin{cases} u_{jk} - u_{jk'}, u_{jk} \geq u_{jk'} \\ 0, u_{jk} < u_{jk'} \end{cases} \tag{4}$$
$$(k, k' \in m; k \neq k'; j \in n)$$

Based on the stochastic dominance relation between the two programs and the value of $\varepsilon_j$, the priority degree function $gj(ak, ak')$ of the program $a_k$ relative to the program $a_{k'}$ for the evaluation dimension $C_j$ is constructed:

$$g_j(a_k, a_{k'}) = \begin{cases} 1, F_{jk}(W)SDF_{jk}(W) \text{ and } u_{jk} \geq u_{jk'} + \varepsilon_j, \\ \frac{u_{jk} - u_{jk'}}{\varepsilon_j}, F_{jk}(W)SDF_{jk}(W) \text{ and } u_{jk} < u_{jk'} + \varepsilon_j, \\ 0, other, \end{cases} \tag{5}$$

$$(k, k' \in m; k \neq k'; j \in n)$$

where $g_j(a_k, a_{k'}) \in [0, 1]$. If the value of $g_j(a_k, a_{k'})$ is larger, it means that the program $a_k$ is better than the program $a_{k'}$.

According to the simple weighting method, the overall priority degree matrix $G = \left[ g_j(a_k, a_{k'}) \right]_{m \times m}$ for the comparison of the pairwise programs can be established, where $g_j(a_k, a_{k'})$ is the overall priority degree of the program $a_k$ relative to the program $a_{k'}$.

$$g(a_k, a_{k'}) = \sum_{j=1}^{n} s_j g_j(a_k, a_{k'}) (k, k' \in m; k \neq k'; j \in n) \tag{6}$$

$g(a_k, a_{k'})$ represents the total credibility that the program $a_k$ is better than the program $a_{k'}$. The larger the value of $g(a_k, a_{k'})$, the greater the degree to which the program $a_k$ is dominant over the program $a_{k'}$.

### 4.3.2. Calculation of the Ranking Value of Each Program

According to the overall priority degree matrix $G$, let $o_k^+$ and $o_k^-$ respectively represent the "outflow" and "inflow" of the program, the calculation formulas of which are as follows.

$$o_k^+ = \sum_{k'=1; k \neq k''}^{m} d_{kk'}^j g(a_k, a_{k'}) \tag{7}$$

In Equation (7), $o_k^+$ represents the total credibility that the program $a_k$ is better than all other programs; the larger the value of $o_k^+$, the better the program $a_k$. Moreover, $o_k^-$ represents the total credibility that the program $a_k$ is inferior to all other programs. The smaller the value of $o_k^-$, the better the program $a_k$.

According to $o_k^+$ and $o_k^-$, the ranking value $o_k$ of program $a_k$ can be calculated as follows.

$$o_k = o_k^+ - o_k^- \tag{8}$$

The larger the value of $o_k$, the better the program $a_k$. The programs can be sorted according to the calculated values of $o_k$, after which the optimal program can be selected.

## 5. Problem-Solving in The Medical Treatment Process Based on Flowing Trimming

Medical and health services are important components of social public services and public affairs management. With the continuous progression of society, patients' consumption consciousness has been awakened. As consumers, the services that patients enjoy cannot satisfy them, and the existing service system and the patient treatment process are no longer suitable for the current situation. Only by solving the problems in the medical treatment process can we better adapt to the new needs present in this new era.

### 5.1. Identifying Problematic Service Touchpoints in the Treatment Process

Through field research and user interviews, the patient treatment process is presented in the form of a user journey map, as shown in Figure 6. In the user journey map, the entire treatment process is divided into three stages: before, during, and after treatment. It is evident that there are four touchpoints in the treatment process in which users have a poor experience. With the help of the internet and information technology, the problems of waiting and examination have been analyzed and solved, while the key touch points that

affect the patient experience, consultation, and queuing for medication, are often selected and often overlooked. In this paper, we use a service flow pruning approach to optimize the probability problem of these two touchpoints.

(1) Consultation. The time for consultation is limited and the doctor's workload is heavy, so the phenomenon of "waiting for three hours, then treatment for three minutes" often occurs. It is a common scenario that a doctor only asks for some basic information, and then prepares the prescription and checklist before the patient has finished his or her explanation. Insufficient communication between doctors and patients will not only cause conflicts, thereby decreasing patients' trust in doctors, but will also affect the treatment experience.

(2) Queuing for medicine. Each patient takes different types and quantities of medication, and the pharmacy dispenses or makes medication at different speeds. Therefore, it is necessary to queue up again to pick up the medicine, i.e., it is necessary for the previous person to finish picking up before the next person, resulting in inefficient service and the irritation of patients.

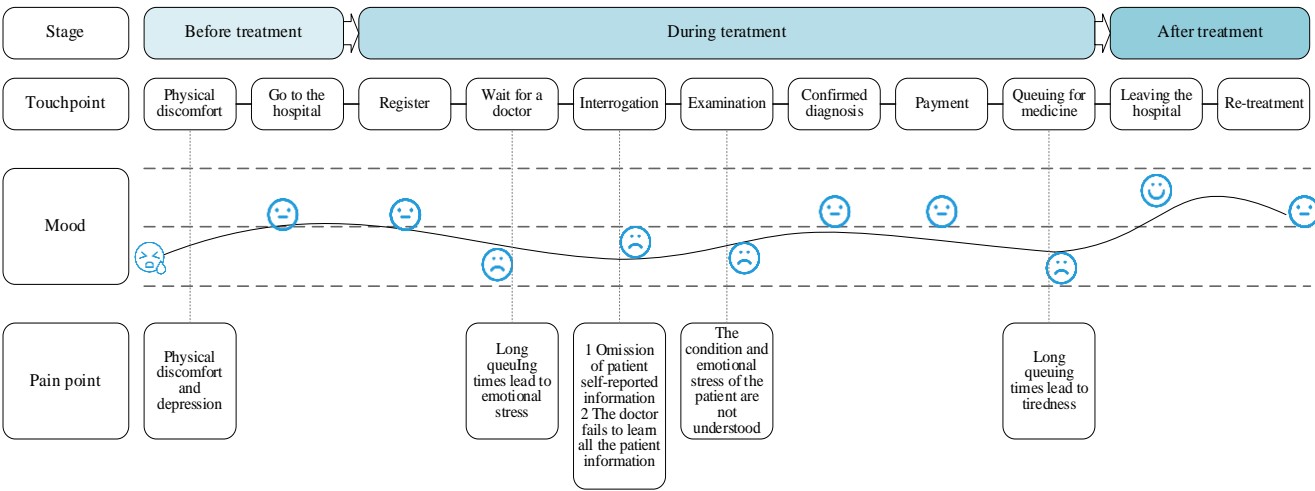

**Figure 6.** The user journey map of the patient treatment process.

*5.2. Flow Analysis of Problematic Service Touchpoints in the Medical Treatment Process*

5.2.1. Stakeholder Identification of Problematic Service Touchpoints Based on the Service Blueprint

As presented in Figure 7, the stakeholders involved in the problematic touchpoints were identified according to the levels of users, stakeholders in the system who have a direct touch with users, stakeholders in the system who do not have a direct touch with users, and support systems/collaborators outside the system. The touchpoint of interrogation includes the patient, doctors, medical records, and the medical record management system. The touchpoint of queuing for medicine includes the patient, hospital pharmacy, hospital, and medicine supplier.

5.2.2. Flow Identification and Flow Model Construction

The types of flows between stakeholders at two adjacent levels were analyzed and judged, and the flow model was established, as shown in Figure 8.

(1) Consultation. The patient's self-reported information given to the doctor is incomplete, causing inefficient flow. The patient information recorded by the doctor is incomplete. The medical record information uploaded to the medical record management system is incomplete. The medical record management system stores and records incomplete patient information. The medical record provides doctors with an incomplete patient medical history and incomplete information that needs to be asked about. The doctor

makes a diagnosis with less information about the patient. All of these situations cause inefficient flow.

(2) Queuing for medicine. Patients provide the hospital pharmacy with prescriptions for required medicines. The hospital pharmacy provides the hospital (management departments) with required medicine information. The hospital and medicine suppliers need to communicate medicine information. Suppliers provide medicines to the hospital—normal flow; the hospital provides medicines to the hospital pharmacy. All of these situations cause normal flow. When the hospital pharmacy provides medicines to patients with low efficiency, this causes a conductivity disadvantage flow.

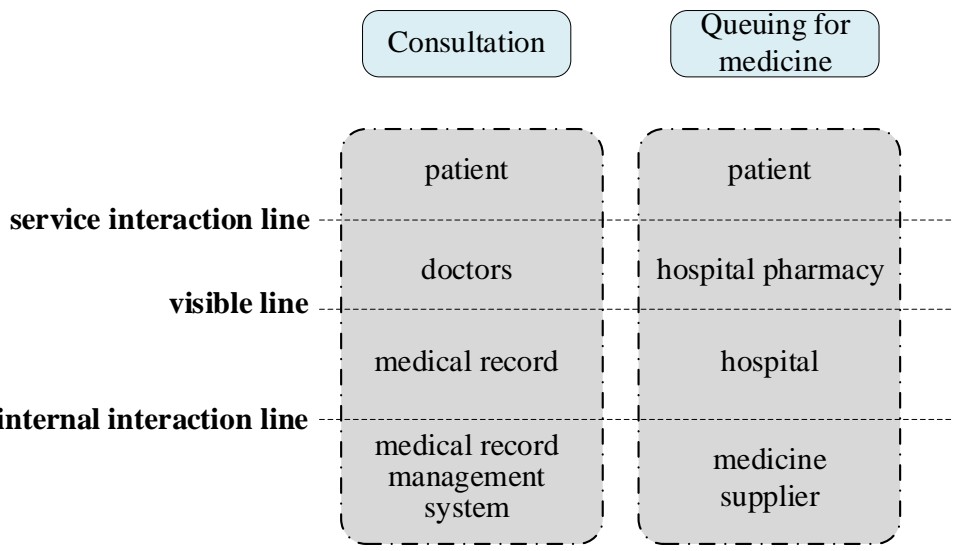

**Figure 7.** The stakeholders in the problematic service touchpoints.

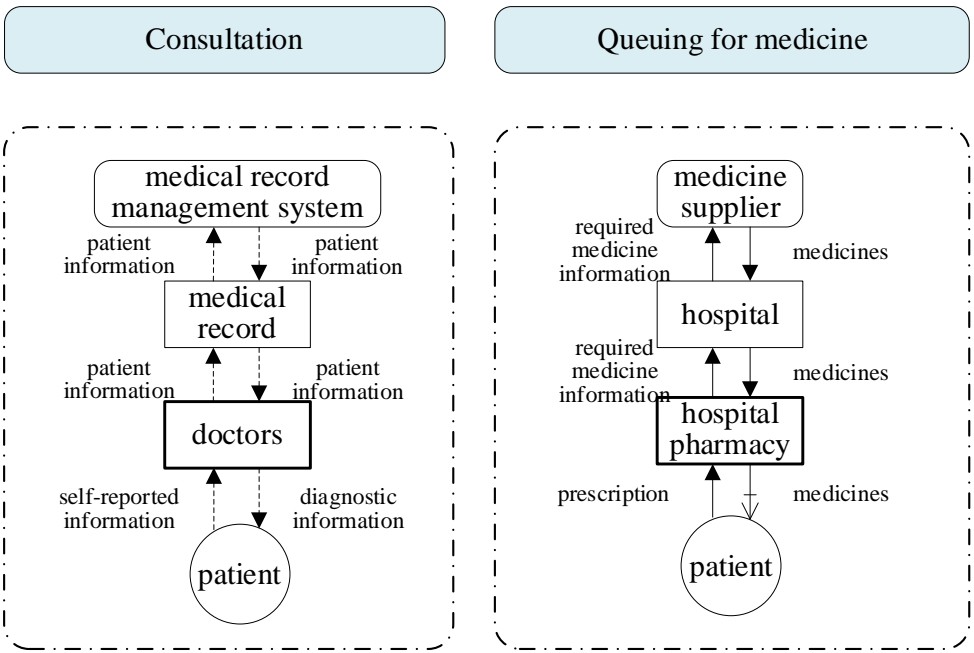

**Figure 8.** The flow model.

5.2.3. Determine the Stream to Be Trimmed

According to the flow model, the top-level flow disadvantages that affect the patient's experience were identified as those to be trimmed.

(1) Consultation. Doctors have a limited time for interrogation, so patients do not have enough time to recall and describe comprehensive medical information, causing inefficient flow.
(2) Queuing for medicine. It takes time for the pharmacy to dispense medicine, which decreases the efficiency of medicine delivery, causing a conductivity disadvantage flow.

### 5.3. Select the Trimming Rule to Perform Trimming

Different trimming rules were attempted to determine the flow to be trimmed.

(1) Consultation

Rule 1: Directly delete the self-reported information flow disadvantage and set a consultation time for each patient so that doctors and patients can fully communicate.
Rule 2: The patient cannot be deleted directly.
Rule 3: Other resources must be found to replace the function of self-reported information flow.
Rule 4: Other resources must be found to replace patients to provide medical information.
Rule 5: Doctors are both the recipients and providers of information in the service process, and they cannot be replaced at present.
Rule 6: Other resources must be found to eliminate the self-reported information flow disadvantage.

(2) Queuing for medicine

Rule 1: The medicine substance flow cannot be deleted directly.
Rule 2: The pharmacy cannot be deleted directly.
Rule 3: Finding other resources to replace the function of the medicine substance flow is beyond the scope of this research.
Rule 4: Other resources must be found to replace the hospital pharmacy to provide medicines.
Rule 5: Other resources must be found to replace patients for receiving medicines.
Rule 6: Other resources must be found to eliminate the low-efficiency medicine delivery flow.

### 5.4. Carry Out Resource Analysis to Achieve a Conceptual Program

(1) Table 2 presents the resource analysis of the problematic service touchpoint of consultation.

**Table 2.** The resource analysis of the problematic service touchpoint of consultation.

| Resource Type | Inside the System | Outside the System |
| --- | --- | --- |
| Material resource | Medical records; medical record management system | Medical records of other hospitals; mobile phones |
| Social resource | Patient; doctors | Doctors in other departments; doctors in other hospitals; other hospitals |
| Information resource | Patient's self-reported information; patient medical record information; doctor's consultation information | Useful information on the internet; consultation information in other hospitals |
| Time resource | Consultation time; time taken for doctors to view medical records; time taken for doctors to fill in medical records | Onset time; waiting time before consultation |
| Space resource | Consultation room | Waiting area |

Rule 3 corresponding program: Mobile phones and the patient's self-reported information can be selected to guide the generation of a specific program based on resource analysis. Patient-side medical records can be generated so that electronic information replaces the function of self-reported information. Patients only need to fill in their informa-

tion and current condition on the patient side of the mobile phone according to the prompts in advance.

Rule 4 corresponding program: Mobile phones and the onset time can be selected to guide the generation of specific programs based on resource analysis. In other words, a health-tracking system can be established to record body information at any time, after which data analysis can be performed and data can be uploaded to the hospital system.

Rule 6 corresponding program: Doctors in other hospitals can be selected to guide the generation of a specific program based on resource analysis. This will increase the number of doctors available for busy departments.

(2)    Table 3 presents the resource analysis of the problematic service touchpoint of queuing for medicine.

**Table 3.** The resource analysis of the problematic service touchpoint of queuing for medicine.

| Resource Type | Inside the System | Outside the System |
| --- | --- | --- |
| Material resource | Medicines | Smart devices; medicines in social pharmacies |
| Social resource | Patient; dispensing doctors; dispensing window staff; medicine suppliers | Social pharmacies; couriers |
| Information resource | Prescriptions | Location information of other pharmacies; storage information of other pharmacies |
| Time resource | Queuing time for medicine; time required to dispense medicine | Other free time |
| Space resource | Hospital pharmacies | Waiting area |

Rule 4 corresponding program: Social pharmacies can be selected to guide the generation of specific programs based on a resource analysis. The hospital can cooperate with various pharmacies in the area. After a diagnosis is completed, the patient can choose to make an appointment to obtain their medicine at a nearby pharmacy according to the prescription.

Rule 5 corresponding program: Couriers can be selected to guide the generation of specific programs based on a resource analysis. The hospital can then cooperate with the couriers. In this way, the patient can leave the hospital after their diagnosis is completed, and the hospital will deliver the medicine to their door via courier.

Rule 6 corresponding program: Smart devices can be selected to guide the generation of specific programs based on resource analysis. In other words, intelligent dispensing robots can be introduced to assist pharmacies to improve their efficiency.

*5.5. Optimal Program Selection*

According to the contents of Section 4.3, four programs were obtained for the problematic service touchpoint of consultation, which are respectively recorded as $a_1$, $a_2$, $a_3$, and $a_4$ in the order of the trimming rules. Moreover, three programs were obtained for the problematic service touchpoint of queuing for medicine, which are respectively recorded as $b_1$, $b_2$, and $b_3$ in the order of the trimming rules. Based on a literature survey, the following five factors that affect the quality of hospital services were identified: advanced equipment ($C_1$), service reliability ($C_2$), information convenience ($C_3$), empathy ($C_4$), and service efficiency ($C_5$). These five influencing factors were considered as the evaluation dimensions of this paper. The weighting system $S = (0.18, 0.27, 0.14, 0.15,$ and $0.2)$ of the five evaluation dimensions was constructed via the expert scoring method. Based on previous research, a questionnaire was set up to allow users who have had hospital experience to score the programs in terms of different evaluation dimensions, and the distribution information of the evaluation values was obtained. To further determine the stochastic dominance

relations between programs, it was necessary to calculate the probability of the evaluation values of programs in different evaluation dimensions. The specific calculation results are reported in Tables 4 and 5.

**Table 4.** The probability distribution of the evaluation values of programs in different evaluation dimensions (problematic service touchpoint: consultation).

| $A_m$ | $a_1$ | | | | | $a_2$ | | | | | $a_3$ | | | | | $a_4$ | | | | |
|---|---|---|---|---|---|---|---|---|---|---|---|---|---|---|---|---|---|---|---|---|
| $S_n$ | 1 | 2 | 3 | 4 | 5 | 1 | 2 | 3 | 4 | 5 | 1 | 2 | 3 | 4 | 5 | 1 | 2 | 3 | 4 | 5 |
| $C_1$ | 0.04 | 0.18 | 0.42 | 0.28 | 0.08 | 0 | 0.08 | 0.26 | 0.50 | 0.16 | 0 | 0.06 | 0.28 | 0.40 | 0.26 | 0.04 | 0.16 | 0.40 | 0.32 | 0.08 |
| $C_2$ | 0 | 0.12 | 0.20 | 0.48 | 0.20 | 0 | 0.12 | 0.10 | 0.38 | 0.40 | 0 | 0.06 | 0.18 | 0.56 | 0.20 | 0 | 0.04 | 0.14 | 0.48 | 0.34 |
| $C_3$ | 0 | 0.02 | 0.46 | 0.34 | 0.18 | 0 | 0.02 | 0.14 | 0.44 | 0.40 | 0 | 0 | 0.26 | 0.40 | 0.34 | 0 | 0.04 | 0.48 | 0.32 | 0.16 |
| $C_4$ | 0 | 0.02 | 0.38 | 0.48 | 0.12 | 0 | 0.08 | 0.32 | 0.44 | 0.16 | 0 | 0.04 | 0.40 | 0.36 | 0.20 | 0 | 0.02 | 0.22 | 0.54 | 0.22 |
| $C_5$ | 0.08 | 0.24 | 0.30 | 0.24 | 0.14 | 0 | 0.06 | 0.10 | 0.44 | 0.40 | 0 | 0.02 | 0.24 | 0.46 | 0.28 | 0 | 0.24 | 0.20 | 0.40 | 0.16 |

**Table 5.** The probability distribution of the evaluation values of programs in different evaluation dimensions (problematic service touchpoint: queuing for medicine).

| $b_m$ | $b_1$ | | | | | $b_2$ | | | | | $b_3$ | | | | |
|---|---|---|---|---|---|---|---|---|---|---|---|---|---|---|---|
| $S_n$ | 1 | 2 | 3 | 4 | 5 | 1 | 2 | 3 | 4 | 5 | 1 | 2 | 3 | 4 | 5 |
| $C_1$ | 0 | 0.06 | 0.28 | 0.42 | 0.24 | 0 | 0.08 | 0.20 | 0.54 | 0.18 | 0 | 0 | 0.14 | 0.50 | 0.36 |
| $C_2$ | 0 | 0.10 | 0.14 | 0.44 | 0.32 | 0 | 0.06 | 0.28 | 0.42 | 0.24 | 0 | 0.18 | 0.30 | 0.34 | 0.18 |
| $C_3$ | 0 | 0.04 | 0.28 | 0.36 | 0.32 | 0 | 0.06 | 0.14 | 0.54 | 0.26 | 0 | 0.08 | 0.40 | 0.26 | 0.26 |
| $C_4$ | 0 | 0.08 | 0.28 | 0.32 | 0.32 | 0 | 0.06 | 0.20 | 0.44 | 0.30 | 0 | 0.10 | 0.46 | 0.26 | 0.18 |
| $C_5$ | 0 | 0.08 | 0.22 | 0.40 | 0.30 | 0 | 0.10 | 0.12 | 0.40 | 0.38 | 0 | 0.02 | 0.26 | 0.38 | 0.34 |

Then, according to the probability distribution information of the evaluation values, the cumulative distribution function for each program in a certain evaluation dimension was obtained. Taking the program $a_1$ in the evaluation dimension $C_1$ as an example, the cumulative distribution function is as follows.

$$F_{11}(w_{11}) = \begin{cases} 0 & w_{11} < 1 \\ 0.04 & 1 \leq w_{11} < 2 \\ 0.22 & 2 \leq w_{11} < 3 \\ 0.64 & 3 \leq w_{11} < 4 \\ 0.92 & 4 \leq w_{11} < 5 \\ 1 & w_{11} \geq 5 \end{cases}$$

According to Equation (1) and the cumulative distribution function, a stochastic dominance relation matrix between the pairwise programs in a certain evaluation dimension was established. Taking the evaluation dimension $C_1$ as an example, the stochastic dominance relation matrix is as follows.

$$H1 = \begin{pmatrix} - & SD & SD & SD \\ - & - & SD & - \\ - & - & - & - \\ - & SD & SD & - \end{pmatrix}$$

According to Equations (2) through (4), the preference thresholds of different evaluation dimensions were calculated. For the problematic service touchpoint of consultation, the preference thresholds of the evaluation dimensions are as follows.

$$\varepsilon 1 = 0.42, \varepsilon 2 = 0.21, \varepsilon 3 = 0.37, \varepsilon 4 = 0.41, \varepsilon 5 = 0.62$$

For the problematic service touchpoint of queuing for medicine, the preference thresholds of the evaluation dimensions are as follows.

$$\varepsilon 1 = 0.27, \varepsilon 2 = 0.31, \varepsilon 3 = 0.2, \varepsilon 4 = 0.31, \varepsilon 5 = 0.09$$

According to Equations (5) and (6), the overall priority degree matrix of each program relative to other programs in the problematic service touchpoint of consultation was calculated as follows.

$$G = \begin{pmatrix} - & 0.790 & 0.722 & 0.561 \\ 0.021 & - & 0.096 & 0.229 \\ 0 & 0.316 & - & 0.360 \\ 0.031 & 0.520 & 0.488 & - \end{pmatrix}$$

The overall priority degree matrix of each program relative to other programs in the problematic service touchpoint of queuing for medicine was calculated as follows.

$$G = \begin{pmatrix} - & 0.276 & 0.380 \\ 0.134 & - & 0.180 \\ 0.560 & 0.604 & - \end{pmatrix}$$

According to Equations (7) and (8), the $o_k^+$, $o_k^-$, and $o_k$ values of each problematic service touchpoint program were calculated, as presented in Table 6.

**Table 6.** The $o_k^+$, $o_k^-$, and $o_k$ values of each program.

| Touchpoint | Program | $o_k^+$ | $o_k^-$ | $o_k$ | Ranking |
|---|---|---|---|---|---|
| Consultation | $a_1$ | 0.052 | 2.073 | −2.021 | 4 |
| | $a_2$ | 1.626 | 0.271 | 1.355 | 1 |
| | $a_3$ | 1.306 | 0.676 | 0.63 | 2 |
| | $a_4$ | 1.15 | 1.039 | 0.111 | 3 |
| Queuing for medicine | $b_1$ | 0.694 | 0.656 | 0.038 | 2 |
| | $b_2$ | 0.88 | 0.311 | 0.566 | 1 |
| | $b_3$ | 0.56 | 1.164 | −0.604 | 3 |

According to the $o_k$ values of the program for each service touchpoint, the optimal program for each service touchpoint was determined.

The best program for the service touchpoint of consultation was found to be $a_2$, which had a large gap with the other programs. Therefore, $a_2$, namely setting patient medical records so that electronic information replaces self-reported information, was determined as the optimal solution. Compared with the existing method of patient self-reported information, in the new program patients can recall their condition with a sufficient amount of time during the waiting period, and then fill in their information and current condition on the patient-side according to the prompts. This can prevent the omission of information, as shown in Figure 9. Furthermore, doctors can grasp more comprehensive information about the patient based on the patient's self-filled medical records, which reduces the time taken by doctors asking for basic information and patients' self-reporting, thereby increasing the efficiency and professionalism of the consultation. The reconstructed consultation system is shown in Figure 10.

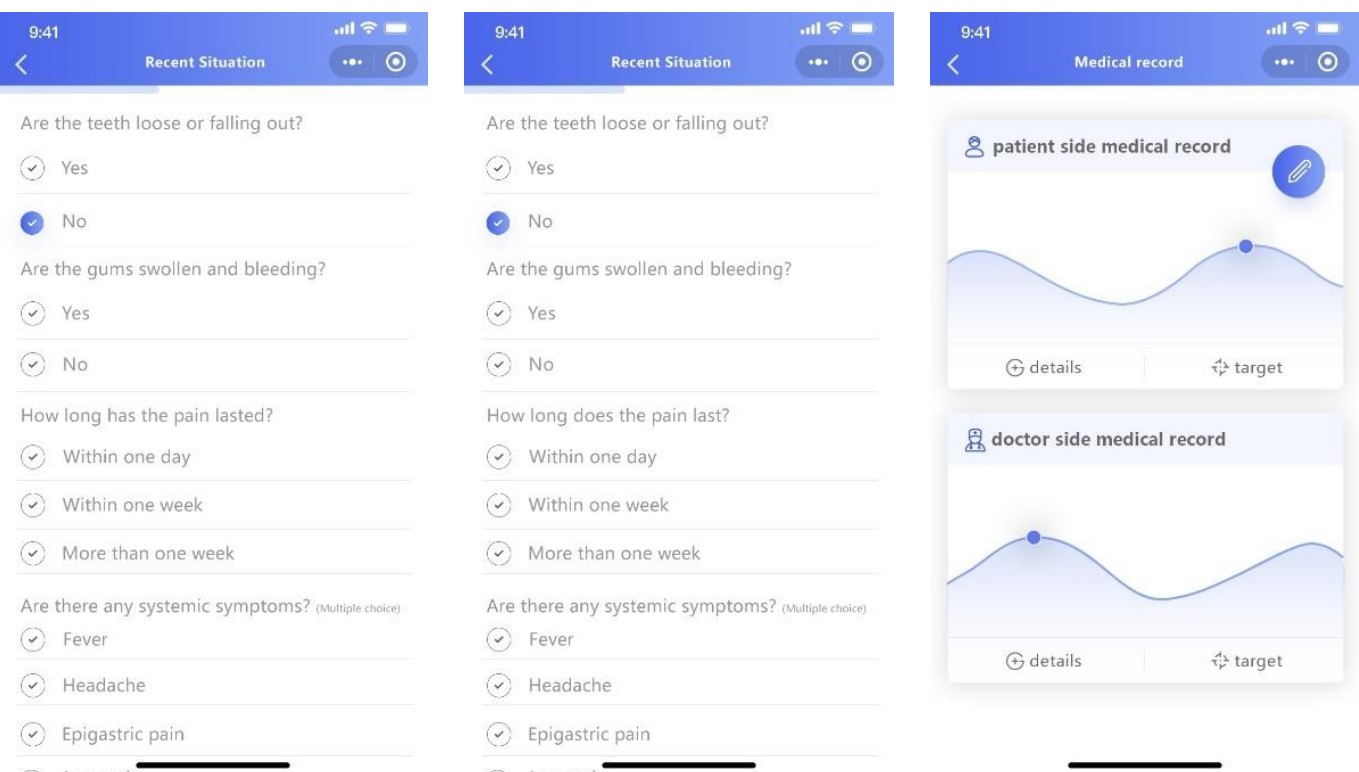

**Figure 9.** A patient's medical record.

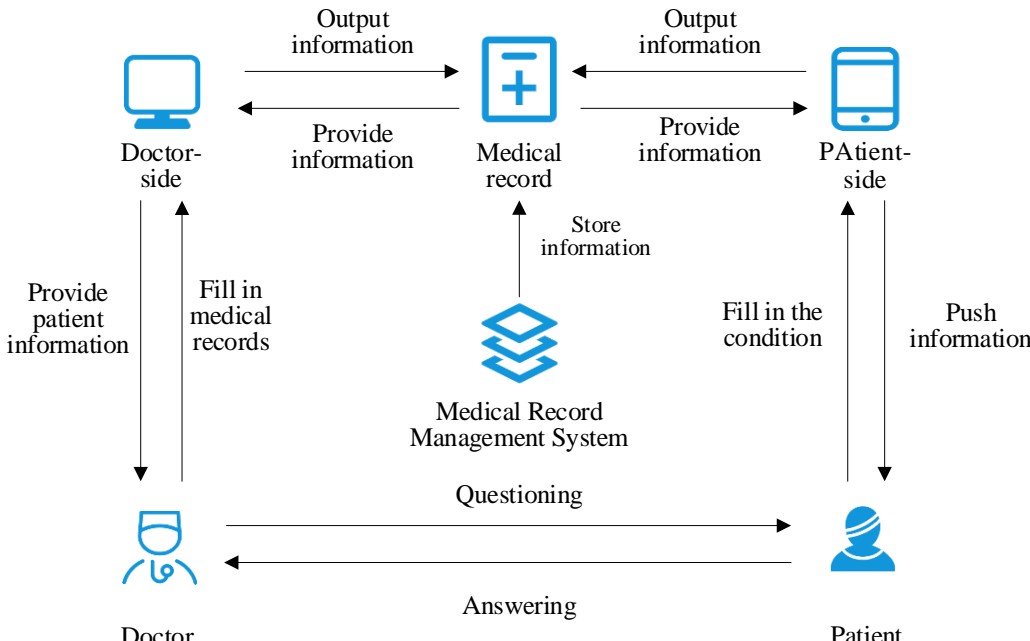

**Figure 10.** The reconstructed consultation system.

The best program for the service touchpoint of queuing for medicine was found to be $b_2$, but the difference between programs $b_1$ and $b_2$ was relatively small. Therefore, programs $b_1$ and $b_2$ were integrated, and appropriate adjustments were made to form an electronic prescription information-sharing system, which would provide more convenient modes for patients to purchase medicines. In the new program, patients would no longer have to queue in front of the hospital pharmacy, and instead, patients can choose to obtain

medicines from the nearest pharmacy or order a courier to their homes via the internet according to their situation, as shown in Figure 11. This model of the separation of medical treatment and medicines basically eliminates the phenomenon of queuing for medicine, which reduces the patient's fatigue. The specific reconstructed medicine acquisition system is shown in Figure 12.

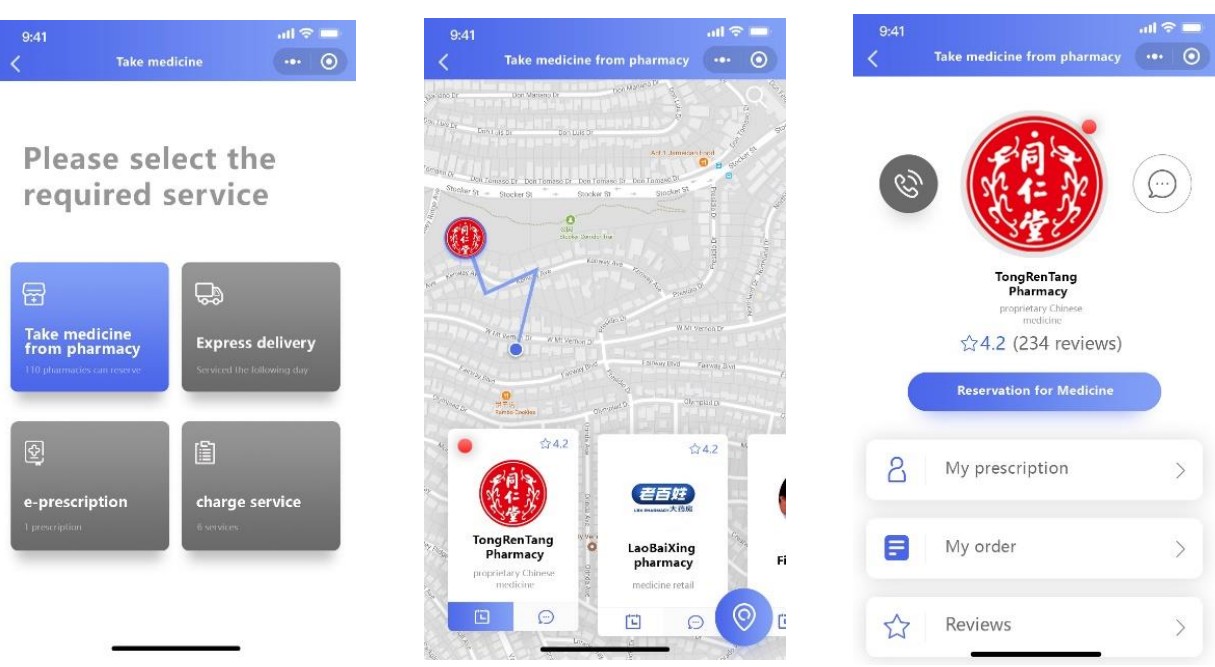

**Figure 11.** An online medicine ordering platform.

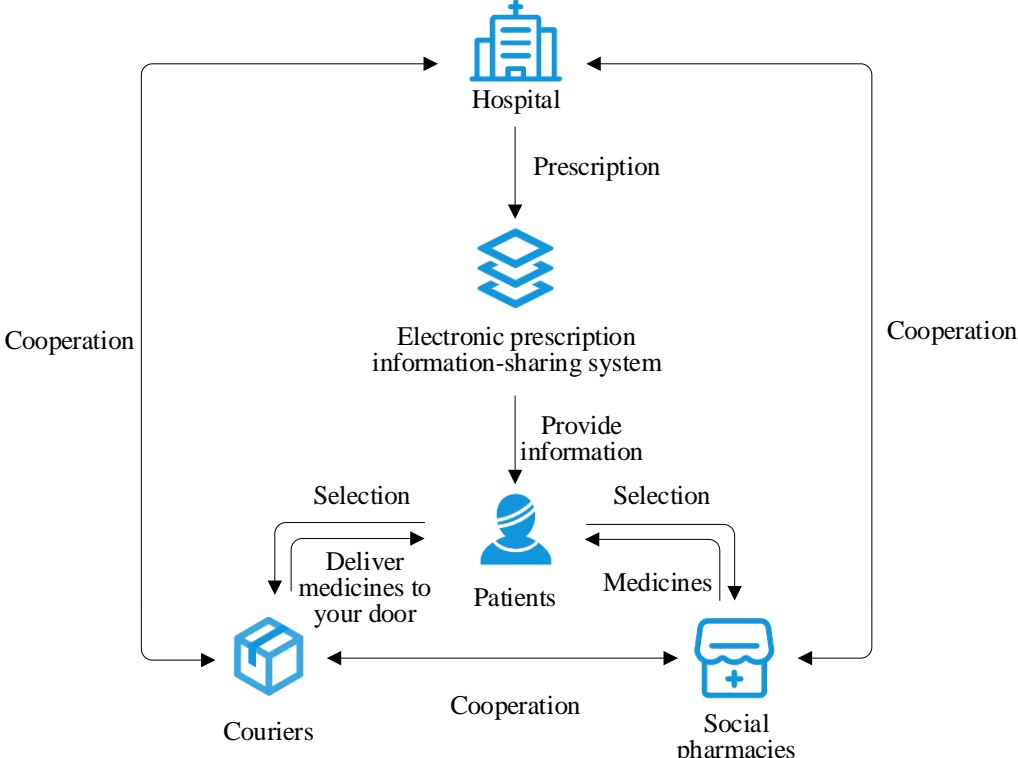

**Figure 12.** The restructured medicine acquisition system.

## 6. Discussion and Conclusions

### 6.1. Discussion

By reading the relevant literature, we learned about the application of various methods in the field of healthcare service process optimization. From the perspective of understanding the application of the methods and the results achieved after application, four methods were selected for analysis, as shown in Table 7. The innovative design process model was analyzed using the medical service process as an example.

**Table 7.** Comparison of methods for service process optimization.

| Author | Method | Result |
|---|---|---|
| Zhang Cheng [52] | Using Lean Six Sigma management system, a control group (conventional) and an observation group (applying Six Sigma) were set up to compare outpatients' consultation, waiting and medication collection time, and satisfaction between the two groups | The observation group (applying Six Sigma) had higher satisfaction in all aspects than the control group (conventional) |
| Li Bin [53] | The model and algorithm of topological sorting in graph theory were applied to a hospital computer information system. Certain patients were selected to be assigned by random number table method and divided into observation group (pre-optimization method) and control group (post-optimization method) | The satisfaction in all aspects was higher in the observation group (optimized method) than in the control group (pre-optimized method) |
| Wu Hongwei [54] | HTCP-net is used to model and optimize the current medical service process of the hospital, and then the internal and external performance indicators of the hospital are obtained through simulation, based on which the medical service process is optimized through process reorganization | Achieved optimization, reorganization, and optimal allocation of resources for medical service processes |
| Adel Hatami-Marbini [55] | Using an optimization method based on mathematical modeling and simulation to determine the best location for an emergency medical center, six scenarios were defined to simulate the model in a dynamic environment, and the survival rate and total cost of each scenario were measured to rank and select the best scenario | Type I and II patients play a critical role in improving survival rates and must be considered when designing EMS facilities, which can help improve survival rates. |

By analyzing and comparing the methods used in the literature in Table 7, Table 8 shows the advantages of the above methods. At the same time, it summarizes the methods proposed in this paper and explains the innovation points. The process defect repair method proposed in this paper comprehensively considers service consumption and service production. Compared with the methods in Table 8, most methods only consider the horizontal analysis of service consumption and do not consider the vertical analysis of service production. In terms of service heterogeneity, this paper chooses the random occupation optimization method to get the optimal solution, which achieves the effect of reducing resource waste and effectively optimizing the medical service process.

The flow clipping and random dominance rule proposed in this paper can solve the problems in the process and meet the needs of users by deleting the flow defects in the system that ultimately led to service contact point problems. This service process trimming method avoids the disadvantages that traditional process trimming methods ignore, namely the characteristics of service processes and focus on production processes. This method analyzes the service process from two perspectives and trims them respectively. The first perspective is the horizontal service process based on service consumption. This is to identify the service contact points that need to be trimmed (inquiry and queuing for medicine). The second perspective is the vertical service process based on service production. These are the flow defects that ultimately lead to service contact point problems in the process of trimming, inquiring, and queuing for medicine. Realized complementary advantages and opened up ideas for solving problems now become apparent, including an

optimized process of consultation and queuing for medicine collection, reduced operating costs, and improved user satisfaction. Considering the heterogeneity of services, which is reflected in the random characteristics of users, the quality of services varies due to the change in location, time, and other factors. We used the random dominant solution to sort and select the optimal solution to make the medical service process economic, efficient, and orderly.

**Table 8.** Several methods are used to innovate the medical service process.

| Method | Brief Introduction | Advantages |
| --- | --- | --- |
| Lean Six Sigma [52], Topological Sorting [53] | Both are tested by making a comparison of observation and control groups for the method, and both rely on management systems and systems as the basis, with models to assist in optimizing the process | Achieve linear time sequencing to reduce operating costs, increase patient satisfaction, speed up processes, improve service quality, and improve input capital efficiency |
| HTCP-net [54], Optimization analysis of mathematical modeling and simulation [55] | All take into account the time aspect of medical assistance to maximize the survival rate and avoid a large and messy medical service process | It can uniformly model the medical service process of different types of patients and has the advantages of considering the real characteristics and improving the efficiency of the service process |

### 6.2. Conclusions

In this paper, the trimming method is applied to the service design field, which not only can solve the problems in service flow for service innovation, but also improves the process trimming method. The flow analysis method, based on the service blueprint and flow defect trimming rules, is proposed for the service flow problem, the trimming-based service flow problem solving process model is established, and the stochastic dominance criterion is applied to the selection process of the optimal service solution so that the optimal solution can be presented. The service process trimming method avoids the drawback that the traditional process trimming method ignores, namely the characteristics of the service process, and focus only on the production process. Examples show that the method can effectively assist designers in solving problems and process innovations in service processes, which is conducive to improving the efficiency of solving service process problems.

This study takes the enhancement of user experiences as the main goal, however, it does not take into account the needs and feelings of all stakeholders. There are three types of streams in the service process: material flow, information flow, and capital flow. This paper conducts vertical service process trimming from the perspective of streams but does not make functional distinctions among the three streams. In the future, the roles of the three streams in the service process can be studied separately to develop trimming rules in a more targeted manner. The concept solving process uses resource analysis, however, this method is limited to a certain extent by the designer's own knowledge base. Therefore, the method of resource analysis can be improved in the future to highly analyze the resources in the service process, explore the hidden resources, pay attention to the organic connection of resources between processes, and reasonably combine configuration and optimization, so as to obtain a more reasonable and optimized solution.

**Author Contributions:** Conceptualization, B.Z., L.S. and Z.X.; methodology, B.Z. and Z.X; investigation, B.Z. and Z.X.; data curation, B.Z., L.S. and Z.X.; writing—original draft preparation, B.Z.; review and editing, B.Z.; funding acquisition, B.Z. All authors have read and agreed to the published version of the manuscript.

**Funding:** This work was supported by the Hebei Social Science Foundation Project [grant number HB21YS041].

**Institutional Review Board Statement:** Not applicable.

**Informed Consent Statement:** Not applicable.

**Data Availability Statement:** Not applicable.

**Conflicts of Interest:** The authors declare no conflict of interest.

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
