# Peer review of "Service Process Problem-Solving Based on Flow Trimming"

_applsci, doi:10.3390/app13042092_

Round 1
Reviewer 1 Report
Dear Authors,
The author needs to clarify the new contribution of the research in the introduction. This part the author did was too sketchy. It is necessary to clearly state the new and motivating points of the article.
The literature review should be placed after the missing section. Authors need to update recent studies. And point out the missing point to carry out this study. The author should have a literature review to compare the results of previous studies conducted in the same research context.
I hope my comments may help you in developing the paper.
Author Response
Thank you for your suggestions, please see the attachment.

Reviewer 2 Report
Thank you for giving me a chance to review the paper. Here are some of my comments:
1) The purpose of the study is stated in the abstract, but the abstract lacks a clear problem statement which normally comes before the purpose of the study.
2) The problem statement and purpose of study are also not clear in the introduction.
3) "Traditional flow analysis lacks practical methods by which to model the elements existing in service touchpoints and systematic methods by which to generate different types of innovative programs. As an important problem-solving tool in the Theory of Inventive Problem-Solving (TRIZ), trimming can effectively optimize system problems, simplify the structure, reduce costs, and achieve innovation [12]". The transition to the use of trimming from TRIZ does not appear to be seamless. The authors should first explain the role of TRIZ in the current paradigm before talking about trimming.
4) In addition to the previous point, it is said by the authors that "trimming can effectively optimize system problems, simplify the structure, reduce costs, and achieve innovation" and that "Trimming was initially applied to the product innovation stage, and as products became more mature, it gradually developed into the field of process innovation". I believe the rationale in using trimming in this study has room for improvement. The authors need to further solidify their justifications on why trimming is the method of emphasis of this study.
5) Although the literature focusses mostly on trimming, the reviewer believes that the authors should introduce a paragraph on the applications of TRIZ first in order to appeal to a larger audience of readers, and improve the readability of the paper. Some recent papers that the authors could cite on TRIZ in general are:
https://doi.org/10.3390/buildings12091456
https://doi.org/10.1016/j.wpi.2022.102155
https://doi.org/10.1007/978-3-031-17288-5_22
https://doi.org/10.3390/app121910105
6) In the conclusion, the problem statement and purpose of study needs to be clearly addressed. Without the problem statement highlighted clearly in the introduction, it makes it difficult to scientifically conclude the paper.
7) The conclusion could also be improved by stating the limitations of the study in a sub-section, and directions for further research in a sub-section.
8) Overall, I believe the paper is well-written, but should undergo formal English editing.
Thank you.
Author Response

(The authors gave the same response as above.)

Reviewer 3 Report
The manuscript is well written and organized. The argument is original and aligned with the scope of the journal. According to my opinion it should be accepted for publication after minor improvements.
The abstract can be reformulated: 1-2 sentences on the context and the need for the study; 1-2 sentences on the methodology; the majority of the abstract on the actual results of the study; 1-2 sentences on key conclusions and recommendations.
What can be the repercussions of the use of the TRIZ "principle" of trimming, applied in this case study, on environmental sustainability? For example, reference is made to the theoretical framework of: Spreafico, C. (2021). Quantifying the advantages of TRIZ in sustainability through life cycle assessment. Journal of Cleaner Production, 303, 126955.
The limitations of the study could be better specified in the conclusions.
Author Response

(The authors gave the same response as above.)
